# Right Ventricular Myocardial Metabolism and Cardiorespiratory Testing in Patients with Idiopathic Pulmonary Arterial Hypertension

**DOI:** 10.3390/diagnostics15192523

**Published:** 2025-10-06

**Authors:** Natalia Goncharova, Aelita Berezina, Daria Ryzhkova, Irina Zlobina, Kirill Lapshin, Anton Ryzhkov, Aryana Malanova, Elizaveta Korobchenko-Andreeva, Olga Moiseeva

**Affiliations:** Almazov National Medical Research Center, Ministry of Health of Russia, Akkuratova Street 2, Saint Petersburg 197341, Russiaryzhkov@almazovcentre.ru (A.R.);

**Keywords:** cardiac PET/CT, [18F]-FDG PET/CT, [13N]-ammonia PET/CT, cardiopulmonary testing, idiopathic pulmonary arterial hypertension, right ventricle

## Abstract

**Background:** Non-invasive diagnostic tools for the assessment of pulmonary arterial hypertension (PAH) are currently being intensively studied. Positron emission tomography (PET)/computed tomography (CT) with [18F]-fluorodeoxyglucose ([18F]-FDG) and [13N]-ammonia is the gold standard for assessing myocardial metabolism and perfusion. The relationship between right ventricle (RV) myocardial metabolism and perfusion and cardiopulmonary exercise testing (CPET) has not been studied. **Objective:** to evaluate correlations between the CPET parameters and RV perfusion and metabolism in IPAH patients. **Methods:** The study comprised 34 IPAH patients (34.2 ± 8.9 years, 4 males, 6 prevalent). Myocardial metabolism and perfusion were assessed using PET/CT with [18F]-FDG and [13N]-ammonia, respectively. CPET, cardiac MRI and invasive hemodynamics were also evaluated. **Results:** Significant negative correlations were registered between [18F]-FDG and [13N]-ammonia uptake by the RV (SUVmax _RV/LV_) and the oxygen consumption, oxygen pulse and positive correlation with the ratio of minute ventilation to carbon dioxide production. The low-risk IPAH patients significantly differed from the intermediate-to-high-risk group in CPET indices and in SUVmax _RV/LV_ metabolism and SUVmax _RV/LV_ perfusion parameters. No reliable differences in CPET indices and [18F]-FDG and [13N]-ammonia uptake by the RV were registered between intermediate- and high-risk patients. **Conclusions:** CPET is a reliable non-invasive diagnostic tool that could distinguish low-risk young IPAH patients without comorbidities from those at intermediate-to-high risk. Significant correlations between CPET parameters and RV myocardial metabolism and perfusion indices, MRI, and invasive hemodynamics confirm the high diagnostic value for CPET.

## 1. Introduction

Right ventricular heart failure is the leading cause of death in patients with pulmonary arterial hypertension (PAH) [1]. The mechanisms of the RV maladaptation closely related to the cardiomyocyte metabolism changes [2]. Adenosine triphosphate deficiency with inefficient oxidative phosphorylation [3] leads to the shift from fatty acid oxidation to an anaerobic pathway of glycolysis in myocardium [4]. Reactive oxygen species and lactate alter the functioning of cellular structures, primarily mitochondria [5]. Thus, a vicious circle of discrepancy arises between the RV energy consumption and the possibility of its provision. Elevated cellular glucose consumption is a marker of increased metabolic demand. PET/CT with [18F]-FDG and [13N]-ammonia are the reference methods for myocardial metabolism and perfusion assessment. [18F]-FDG uptake in the RV myocardium has been tightly associated with the severity of PAH [4,6,7]. The high cost and the radiation exposure determine infrequent use of radionuclide methods for maladaptive remodeling assessment in patients with IPAH.

CPET is of great interest as it evaluates the adequacy of oxygen delivery and utilization by tissues [8], identifies the “suffering” system (cardiovascular, respiratory or muscular) and the mechanisms of compensation [9]. CPET indices closely correlate with hemodynamics and the right heart remodeling in patients with IPAH [10,11]. CPET is a perspective non-invasive diagnostic method for the risk assessment during follow-up in addition to the established ESC/ERS 2022 four-strata tools (PAH functional class (WHO), six minute walk test (6MWT) distance and N-terminal pro-brain-type natriuretic peptide (NT-proBNP)) [12]. No data regarding CPET and the RV metabolic and perfusion indices currently exist in IPAH patients.

We hypothesize that there is a close correlation between the RV myocardial metabolism and perfusion and CPET parameters depending on the current concept of PAH severity assessment.

The study aimed to evaluate the correlation between the CPET parameters and RV myocardial perfusion and metabolism using PET/CT with [13N]-ammonia and [18F]-fluorodeoxyglucose ([18F]-FDG) in IPAH patients. The relationships between the CPET parameters and the cardiac MRI, invasive hemodynamics were analyzed for the integrity of the study hypothesis.

## 2. Methods

### 2.1. Data Collection

According to the 2020 national guidelines for the management of pulmonary hypertension [13], the following hemodynamic criteria for PAH were used in the study: mean pulmonary artery pressure (PAP) ≥25 mmHg, pulmonary capillary wedge pressure (PCWP) < 15 mmHg, and pulmonary vascular resistance (PVR) ≥3 Wood units. Pulmonary arterial compliance (PAC) was calculated using invasive hemodynamic data obtained with the right heart catheterization (RHC), as follows: stroke volume/(systolic PAP-diastolic PAP) [14,15]. The exclusion criteria for the study were as follows: PAH, etiologies other than IPAH, moderate to severe lung disease (forced vital capacity <70%; forced expiratory volume in one second <80%; diffusion capacity of the lungs for carbon monoxide <40%), left heart disease, established malignancies, diabetes mellitus, thyroid disease, severe kidney or liver dysfunction, inflammatory diseases, and mental disorders.

All patients underwent standard evaluation with symptoms and demographics assessment; exercise tolerance, via 6MWT distance and CPET; RHC and N-terminal pro-brain-type natriuretic peptide (NT-proBNP); C-reactive protein (CRP); hemoglobin; and estimated glomerular filtration rate (eGFR), calculated according to the CKD-EPI equation. PET/CT with two radiopharmaceuticals, [13N]-ammonia and [18F]-fluordeoxiglucose ([18F]-FDG), was performed no later than 1 month after the RHC. Cardiac MRI was undertaken in all patients using a MAGNETOM Trio A Tim System 3 Tesla (Siemens AG, Erlangen Germany).

All patients underwent cardiopulmonary exercise testing (CPET) on a bicycle ergometer (Ebike, GE HealthCare, Milwaukee, WI, USA) with an increasing load of 10 W/min (RAMP protocol) until they achieved maximum tolerated exertion limited by symptoms. Gas exchange was measured by the respiratory cycle method using a calibrated Oxycon Pro device (Cardinal Health, Halberstadt, Germany).

ESC/ERS 2022 3-strata risk stratification was performed in all patients using an online calculator [https://www.pahinitiative.com/hcp/risk-assessment/calculators] (URL accessed on 14 January 2025).

### 2.2. PET/CT Protocol

Cardiac PET/CT («Discovery 710», GE Healthcare, Waukesha, WI, USA) with [18F]-FDG and [13N]-ammonia was performed in all patients in two separate days. Myocardial glucose metabolism was assessed by [18F]-FDG PET/CT. Patients fasted at least 6 h before the [18F]-FDG PET/CT procedure. The hyperinsulinemic euglycemic clamp was performed as described in the publication by R. A. De Fronzo (1979) [16]. A polyethylene cannula was inserted into the cubital vein to infuse insulin and 10% glucose solution through a three-way stopcock. Insulin was administrated at a constant rate of 40 mIU/min/m^2^. The rate of intravenous glucose infusion was adjusted manually. Glucose level in the blood was checked every 5 min. Stable euglycemia was considered to be achieved if three consecutive blood glucose measurements differed from each other within ±5%. Stable euglycemia was usually achieved 1 h after the start of insulin and glucose infusion. A standard dose of 5 MBq/kg (<550 MBq) [18F]-FDG was administered intravenously, when stable glycemia was established (optimal glycemia was 5 mmol/L). Cardiac PET/CT scans were acquired 40 min after [18F]-FDG intravenous administration in a static mode. Low dose CT scan was performed immediately prior to PET for attenuation correction. PET perfusion scanning was performed at rest, 5 min after intravenous administration of 10 MBq/kg [13N]-ammonia in a static mode immediately after low dose CT transmission.

The regions of interest (ROIs) of the left ventricle (LV) and RV myocardial uptake were identified visually. The regions of interest (ROIs) were drawn on the RV free wall, and LV lateral wall on the static transaxial images in order to measure the LV and RV myocardial uptake of both [18F]-FDG and [13N]-ammonia [7]. The maximal standardized uptake values of radiopharmaceuticals (SUVmax) were recorded in each ROI using AWS 4.6 software (GE Healthcare, USA). The right ventricular myocardium metabolism (SUVmax _RV/LV_ metabolism) and perfusion (SUVmax _RV/LV_ perfusion) were calculated as the ratio of SUVmax RV free wall and SUVmax LV lateral wall for the [18F]-FDG and [13N]-ammonia PET images, respectively. To assess the relationship between perfusion and metabolism in the right heart myocardium, the ratio of metabolism SUVmax _RV/LV_ to perfusion SUVmax _RV/LV_ was calculated.

The study was conducted in accordance with the Declaration of Helsinki and approved by the ethics committee of the center (protocol N 04-23, approved on 17 April 2023). Informed consent was obtained from all subjects participating in the study. Written informed consent was obtained from all patients for publication in the article.

### 2.3. Study Population

The study population comprised 34 adult (18–52 years old) Caucasian IPAH patients prospectively recruited in a single-PH referral center between February 2020 and October 2024. Patients were divided into three groups depending on ESC/ERS 2022 risk status: low risk *(n*=11), intermediate risk (*n* = 17) and high risk (*n* = 6) (Table 1). Twenty-eight incident patients underwent initial PAH diagnostics and were treatment naïve. Six prevalent patients were enrolled into the study within 6 to 12 months from the initial diagnosis of IPAH in the center. Three prevalent IPAH patients with positive acute vasoreactive testing (AVT) received calcium channel blockers (CCB). Two III–IV functional class (FC) (WHO) patients received triple combination therapy, including riociquate, sildenafil, macitentan, ambrisentan, iloprost and selexipag. One III FC patient received sildenafil and ambrisentan. Prevalent patients received CCB or PAH-specific therapy for at least three months after the diagnosis of IPAH was established. All procedures, including RHC, MRI and PET/CT, were undertaken on a background PAH therapy in prevalent patients. Acute vasoreactive testing was performed as a part of first follow-up assessment of CCB therapy efficacy in AVT positive prevalent patients according to the guideline [14,15]. Three-strata ESC/ERS 2022 risk stratification was performed in prevalent patients using all parameters obtained at the time of the study conduction. Survival data were not provided, as no deaths have been reported to date.

### 2.4. Statistical Analysis

Demography, PAH FC (WHO), smoking status, invasive hemodynamics, NT-proBNP, hemoglobin, CRP and eGFR, CPET, MRI parameters and PET/CT indices were compared in three risk groups. Variables were tested for normality, and numerical parameters with a normal distribution were presented as mean ± standard deviation (M ± SD), while numerical parameters with an abnormal distribution were presented as median and interquartile range (IQR; M ± 25%, 75%). Categorical variables were presented as absolute numbers and percentages and compared using Fisher’s exact test, as appropriate. The mean values of two groups were compared using the *t*-test and ANOVA test for comparison of means of three groups, as appropriate. Pearson’s correlation analysis was used to assess the direction and statistical strength between continuous variables. A statistically significant difference was determined with a two-tailed *p* value less than 0.05. We evaluated cardiac MRI and PET/CT parameters using univariate and multivariate linear regression analysis to determine the independent predictors for the peak oxygen (O_2_) consumption and minute ventilation per unit carbon dioxide production. The statistical analyses of the data were conducted with Statistica for Windows, version 10.0 (StatSoft: Tulsa, OK, USA).

## 3. Results

### 3.1. Patients’ Characteristics Depending on ESC/ERS Risk Status

Patients of the three ESC/ERS 2022 risk groups did not differ in age, body mass index, smoking, eGFR, or hemoglobin level. Low-risk patients differed significantly from the intermediate-to-high risk population in terms of exercise tolerance, hemodynamics and right heart remodeling, NT-proBNP level and RV myocardial metabolism, and perfusion. No significant difference was revealed between intermediate- and high-risk patients in FC (WHO), 6MWT distance and CPET parameters, cardiac MRI and hemodynamics (mean PAP, right atrial pressure, PCWP, PVR), arterial oxygen saturation, and RV myocardial metabolism and perfusion (Table 1).

No significant correlations were found between age and hemodynamics, RV ESV index, RV EF and LV SV index obtained using cardiac MRI, and PET/CT and CPET parameters (Appendix A). No significant differences were registered between prevalent (n = 6) and incident patients in [18F]-FDG SUVmax _RV/LV_ lateral wall (0.924 ± 0.476 vs. 0.825 ± 0.348, *p* = 0.5), [13N]-NH3 SUVmax _RV/LV_ lateral wall (0.825 ± 0.176 vs. 0.815 ± 0.157, *p* = 0.8) and SUVmax 18F-FDG/SUVmax [13N]-NH3 _RV/LV_ lateral wall (1.699 ± 0.967 vs. 1.381 ± 0.763, *p* = 0.4). Similar values of VO_2_ peak (12 [11.1; 18] vs. 14.5 [11.1; 17.9] mL/min/kg, *p* = 0.9), VO_2_ peak predicted (54.2 ± 31 vs. 54.5 ± 16.9%, *p* = 0.9) and VE/VCO_2_ (44.6 ± 9.6 vs. 49.7 ± 13.7, *p* = 0.4) were observed in prevalent and incident patients.

### 3.2. CPET Parameters According to the ESC/ERS 2022 Risk Status

Low-risk patients differed significantly in predicted workload, peak oxygen consumption in absolute (VO_2_ peak) and predicted values (%VO_2_ peak predicted), the ratio between oxygen consumption and workload (ΔVO_2_/ΔWR), and heart rate slope (HR/Vkg) when compared with the intermediate-to-high-risk patients, whereas no significant difference was observed between the intermediate- and high-risk groups (Table 1). No significant difference was registered in ventilatory equivalents, exercise desaturation and predicted breathing reserve (% BR predicted) between the three risk groups. Low-risk patients had significantly lower ventilatory equivalent for carbon dioxide on anaerobic threshold (VE/VCO_2_ AT) when compared with the intermediate- and high-risk patients.

### 3.3. Correlations Between CPET and Hemodynamics

Significant inverse correlations were identified between the mPAP and the workload, peak O_2_ consumption, oxygen pulse, and PetCO_2_, and a direct correlation was observed with VE/VCO_2_ (Table 2). No significant correlations were registered between mean RAP and CPET parameters. Significant positive correlations were recorded between cardiac index and the workload, VO_2_ peak, and oxygen pulse, and an inverse correlation was observed with the heart rate slope (HR/Vkg), VE/VCO_2_ and anaerobic work efficiency (ΔVO/ΔWR). A positive correlation was registered between the cardiac index and the PetCO_2_. A direct correlation was recorded between PAC and the predicted load, O_2_ consumption, oxygen pulse and PetCO_2_.

### 3.4. Correlations Between CPET and Cardiac MRI Parameters

We assessed the relationship between CPET parameters with established MRI determinants of prognosis in PAH patients, as follows: right ventricular end–systolic volume index (RV ESV index), left ventricular stroke volume index (LV SV index), and right ventricular ejection fraction (RV EF). In addition, we assessed the ratio of the RV ESV index/LV ESV index (Table 3, Figure 1).

A significant inverse correlation was registered between RV ESV index and the predicted workload, O_2_ consumption at peak of exertion, and at anaerobic threshold, PetCO_2_.

A significant positive correlation was found between the LV stroke volume index and predicted load, O_2_ consumption at peak of exertion and at anaerobic threshold, oxygen pulse, PetCO_2_ peak, and dO_2_/dW and an inverse correlation was observed with VE/VCO_2_.

No reliable correlations were found between the RV ejection fraction and the CPET indices.

The largest number of correlations were observed between the CPET parameters identified with the ratio of the RVESVindex/LVESVindex. Significant negative correlations were found between the ratio of the RVESVindex/LVESVindex and predicted workload, O_2_ consumption, oxygen pulse, PetCO_2_ and dO_2_/dW. A direct correlation was recorded between the RVESVindex/LVESVindex and VE/VCO_2_.

### 3.5. Correlations Between CPET Parameters and [18F]-FDG and [13N]-Ammonia RV Myocardial Uptake

Significant inverse correlations were registered between the SUVmax _RV/LV_ metabolism and O_2_ consumption and PetCO_2_. Direct significant correlations were recorded between the RV/LV metabolism and VE/VCO_2_.

Significant inverse correlations were recorded between the values of the SUVmax _RV/LV_ perfusion and workload, O_2_ consumption and PetCO_2_. Direct correlations were recorded between the value of the SUVmax _RV/LV_ perfusion and the magnitude of dead space ventilation increase during exercise, heart rate slope and VE/CO_2_. The inverse correlation was observed between PetCO_2_ AT and the SUVmax _RV/LV_ perfusion value.

A direct significant correlation was found between the ratio of metabolism to perfusion of the RV/LV myocardium and heart rate slope, and negative correlations were observed with workload and O_2_ consumption (Figure 1, Table 4).

### 3.6. Changes in [18F]-FDG and [13N]-Ammonia Uptake by the RV Myocardium Depending on Risk Categories of the Main CPET Determinants of Prognosis

The values of [18F]-FDG SUVmax _RV/LV_ lateral wall and [13N]-NH3-ammonia SUVmax _RV/LV_ lateral wall have not been validated in terms of the mortality risk scores in patients with PAH. Therefore, we divided patients into three risk groups for VO_2_ peak, %VO_2_ peak predicted, and VE/VCO_2_ according to the established ESC/ERS 2022 risk categories [14]. Given the small number of patients in the intermediate risk category for VO_2_ peak and VE/VCO_2_ (*n* = 4), intermediate- and high-risk patients were pooled and compared with low-risk patients (shown in Figure 2) (Table 5).

### 3.7. Peak Oxygen Consumption Category

Significant differences in [18F]-FDG and [13N]-ammonia uptake by the RV myocardium were observed between patients in the low and intermediate-to-high risk VO_2_ peak categories. No significant differences in perfusion and the ratio of metabolism to perfusion of the RV/LV myocardium were registered between the intermediate-risk and high-risk VO_2_ peak categories.

### 3.8. The Percent of Predicted Peak Oxygen Consumption Category

Significant differences in the SUVmax _RV/LV_ metabolism were recorded between patients in the low and high-risk VO_2_ peak % predicted categories, as well as between intermediate- and high-risk categories, while the SUV max _RV/LV_ metabolism did not differ significantly between patients in the low and intermediate-risk VO_2_ peak % predicted category.

The SUV max _RV/LV_ perfusion differed significantly across all risk groups in the VO_2_ peak % predicted category.

No significant differences in the ratio of metabolism to perfusion of the RV/LV myocardium were found depending on the risk category of the peak VO_2_% predicted (Table 5).

### 3.9. The Ratio of Minute Ventilation to Carbon Dioxide Production Category

There was a significant difference in SUVmax _RV/LV_ metabolism and SUV max _RV/LV_ perfusion between low and high-risk VE/VCO_2_ category patients. The ratio of metabolism to perfusion of the RV/LV myocardium did not differ significantly between VE/VCO_2_ risk categories (Table 5).

### 3.10. The Mean Values of [18F]-FDG and [13N]-Ammonia Uptake by the RV Myocardium Depending on the Risk Status and CPET Categories

The mean value of SUVmax _RV/LV_ for [18F]-FDG PET/CT was 0.581 ± 0.39 in the low-risk group and ranged from 0.59 to 0.6 depending on prognostic parameters of CPET (peak VO_2_, peakVO_2_% predicted, VE/VCO_2_). The mean value of SUVmax _RV/LV_ for [18F]-FDG PET/CT ranged from 0.996 ± 0.27 in the intermediate-to-high ESC/ERS 2022 risk cohort and was within the same limits while using intermediate-to-high risk categories for peak VO_2_, peakVO_2_% predicted, and VE/VCO_2_.

The mean value of SUVmax _RV/LV_ for [13N]-ammonia PET/CT was 0.758 ± 0.153 and 0.815–0.862 in the low-risk ESC/ERS 2022 group and in the low-risk category of CPET indicators, respectively. The mean value of SUVmax_RV/LV_ for [13N]-ammonia PET/CT ranged around 0.852 ± 0.151 in the intermediate-to-high ESC/ERS 2022 risk cohort and was within the same limits while using the intermediate-to-high risk categories of peak VO_2_, peak VO_2_% predicted, and VE/VCO_2_.

The ratio of metabolism to perfusion of the RV/LV myocardium was 0.786 ± 0.321 in low-risk patients according to the ESC/ERS 2022 scale and varied, depending on the CPET parameters, from 1.08 to 1.2. The mean value of the ratio of metabolism to perfusion of the RV/LV myocardium was in the range of 1.797 ± 0.751 in patients with intermediate-to-high risk of the ESC/ERS 2022 scale and for CPET indicators of −1.6–2.0.

Thus, the mean values of SUVmax _RV/LV_ for [18F]-FDG and [13N]-ammonia were within the same limits when using ESC/ERS 2022 risk scale or CPET indicators alone (Figure 2 and Figure 3).

### 3.11. PET/CT and MRI Predictores for the Peak Oxygen Consumption and Minute Ventilation per Unit Carbon Dioxide Production

Peak O_2_ consumption was significantly associated with myocardial RV uptake with [18F]-FDG, the ratio of metabolism to perfusion of RV, RV ejection fraction, the ratio of indices RV ESV/LV ESV and LV stroke volume index, according to the univariate regression analysis. Myocardial RV uptake with [18F]-FDG was the independent predictor of peak O_2_ consumption according to the multivariate linear regression analysis (OR-6.5, 95% CI (−11.5–1.25), *p* = 0.016). Minute ventilation per unit carbon dioxide production was significantly associated with the RV metabolism and perfusion according to the univariate regression analysis, but no independent predictor was determined with miltivriate analysis (Table 6).

## 4. Discussion

For the first time, the relationship between CPET and RV myocardial metabolism and perfusion indices has been presented in patients with IPAH without comorbidities.

Cardiac output increase in response to metabolic demand is one of the main determinants of exercise capacity. Pulmonary microvasculature recruitment and PVR decrease provide adequate cardiac output and oxygen delivery to tissues during physical exertion under physiological conditions [17]. Pulmonary arterial pressure and PVR elevation accompanied by RV hypertrophy with subsequent RV dilation and decreased myocardial contractility leads to RV–PA uncoupling. [18,19]. LV underfilling and low stroke volume invariably result in low tissue perfusion, compensatory high tissue oxygen extraction and a metabolic shift towards anaerobic oxidation [8]. Hyperventilation with PETCO_2_ decrease, elevation of VE/VCO_2_ and dead space ventilation are the typical features of respiratory compensation in patients with IPAH [20].

The present study confirmed the close correlations between hemodynamics, RV myocardial remodeling and CPET parameters, and this is consistent with previously published studies [8,20,21]. We did not find any significant correlations between CPET parameters and RV ejection fraction, which might be due to the small patient population and a wide range of intermediate-risk limits for the RV ejection fraction [13,22,23,24,25]. The largest number of correlations between the CPET parameters were identified with the ratio of the RVESV index/LVESV index. The RV/LV volumes ratio was not included in the risk scale stratification [14], though this parameter reliably reflects the severity of RV remodeling and the LV compression in patients with PAH.

High glucose consumption and an early transition to the anaerobic pathway of glycolysis were determined in myocardium in a pressure overload state [25,26]. A close correlation between the RV myocardial metabolism and perfusion parameters and mean PAP, PVR, cardiac output, and RV remodeling was demonstrated in patients with PAH [7,27]. Changes in RV myocardial metabolism were observed even in a slight mPAP and PVR elevation in patients with IPAH. Increased metabolic demand was compensated by perfusion augmentation of the RV myocardium in low-risk IPAH patients with mild PVR elevation and slight RV myocardial remodeling [7]. However, the RV metabolism-to-perfusion ratio dramatically increased in intermediate- and high-risk IPAH patients with severe hemodynamic changes and RV remodeling. In the present study, the rise of [18F]-FDG and [13N]-ammonia uptake by the RV myocardium was accompanied with a decrease in workload, VO_2_ peak, oxygen delivery and an increase of the ratio of minute ventilation to carbon dioxide production depending on the risk status.

The identification of low-risk IPAH patients is the most important task according to the current concept of PAH management. Obtaining a true low-risk profile of an IPAH patient using RV molecular imaging is of paramount importance. Data regarding changes of [13N]-ammonia and [18F]-FDG uptake by RV myocardium, depending on the risk status and the severity of PAH, are scarce [7,28], while the prognostic values of the main CPET indicators are well defined within the ESC/ERS 2022 risk scale [12,13,14]. Thus, we assessed the mean values of [18F]-FDG and [13N]-ammonia uptake by RV myocardium in terms of ESC/ERS risk categories and separately in terms of established prognostic CPET indices (peak VO_2_, peak VO_2_% predicted, VE/VCO_2_). The values of the RV myocardium metabolism and perfusion differed significantly between low- and intermediate-to-high risk IPAH patients in terms of different CPET parameters. Whereas, the intermediate- and high-risk groups did not exhibit significant differences in CPET and PET/CT data, which might be due to the small sample of high-risk patients (n = 6). The RV metabolism-to-perfusion ratio was numerically lower in patients with low-risk CPET indicators in comparison with those with intermediate-to-high risk. Similar values of the RV myocardium [18F]-FDG and [13N]-ammonia uptake were observed in between the risk categories derived from the ESC/ERS 2022 scale and the risk categories of the main CPET indicators.

The mean values of [18F]-FDG and [13N]-ammonia RV myocardial uptake were another topic of interest. The higher SUVmax _RV/LV_ metabolism value was recorded in low-risk patients in the present study, which stands in comparison with our previously published data (0.581 ± 0.39 vs. 0.38 ± 0.09) in patients with IPAH [7]. The SUVmax _RV/LV_ perfusion was only slightly higher in low-risk patients in the present study when compared with those in the previously published study (0.74 ± 0.15 vs. 0.65 ± 0.06), and did not differ in intermediate-risk (0.86 ± 0.14 vs. 0.85 ± 0.15) and high-risk patients (0.79 ± 0.2 vs. 0.79 ± 0.18) between studies. A higher ratio of metabolism to perfusion of the RV/LV myocardium was recorded in the present study compared with the previous one, especially in the low-risk group (0.78 ± 0.32 vs. 0.69 ± 0.27). The differences in mean values of SUVmax _RV/LV_ metabolism and perfusion in low-risk patients between the two studies were likely due to a larger number of low-risk patients (n = 11 vs. n = 6), more severe hemodynamic signs of pulmonary vascular disease (PVR 5.5 [3.9; 7.2] vs. 3.9 [3.1; 5.4] WU; mPAP 40.8 ± 3.9 vs. 30.8 ± 8.3 mm Hg) and more pronounced RV myocardial remodeling (RV ESVi 43.9 ± 12.3 vs. 36.0 ± 8.0 mL/m^2^). The obtained results demonstrate the high heterogeneity of the SUVmax _RV/LV_ metabolism in patients within the low-risk group. PET/CT studies have not been implemented into clinical practice for PAH patients yet, due to their unclear practical significance. Our observation needs further data accumulation and analysis in terms of the interconnection between RV metabolism and perfusion and hemodynamics, cardiac remodeling and outcome in IPAH patients. No specific drugs that target RV myocardium in PAH currently exist. The approved PAH-specific medicines realize positive remodeling and metabolic effect on the RV through mPAP decrease, which has been confirmed by R. Kazimierczyk et al. (2023) [28]. RV metabolism molecular imaging using PET/CT might be essential for the formation of new treatment goals and the creation of new drugs.

We have clearly demonstrated significant correlations between CPET parameters and hemodynamics, and cardiac MRI, consistent with previous studies [10,29,30]. The established interconnection between CPET parameters and RV metabolism is of paramount importance, as the molecular changes preclude clinical manifestation and are tightly connected with hemodynamics in IPAH patients [22,25,27]. The initial strategy and the first six months of IPAH patient management might determine the course of the disease [31]. We suggest that CPET might be an excellent non-invasive diagnostic tool for low-risk status confirmation in young IPAH patients without comorbidities.

## 5. Conclusions

Right ventricular [18F]-FDG myocardial uptake was found to be an independent predictor of peak oxygen consumption in IPAH patients without comorbidities.

CPET has proven to be a reliable non-invasive tool for the differentiation of IPAH patients without comorbidities with a low-risk status and those with an intermediate-to-high risk status.

The accumulation of data on the right ventricular myocardial metabolism in comparison with traditional invasive and non-invasive methods for PAH severity assessment requires prospective validation in a larger cohort of patients with IPAH.

### Limitations

The sample size was small (*n* = 34), with only six patients in the high-risk group, potentially affecting statistical power (e.g., the lack of significant differences between intermediate- and high-risk groups).

The prognostic value of PET/CT and CPET parameters could not be assessed due to an insufficient observation period.

## Figures and Tables

**Figure 1 diagnostics-15-02523-f001:**
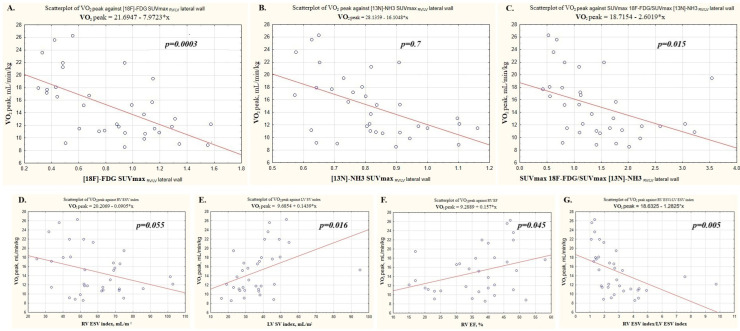
Analysis of factors associated with the aerobic exercise capacity. (**A**) [18F]-FDG SUVmax _RV/LV_ lateral wall; (**B**) [13N]-NH3 SUVmax _RV/LV_ lateral wall; (**C**) SUVmax 18F-FDG/SUVmax [13N]-NH3 _RV/LV_ lateral wall; (**D**) RV ESV index; (**E)** LV SV index; (**F**) RV EF; (**G**) RV ESV index/LV ESV index.

**Figure 2 diagnostics-15-02523-f002:**
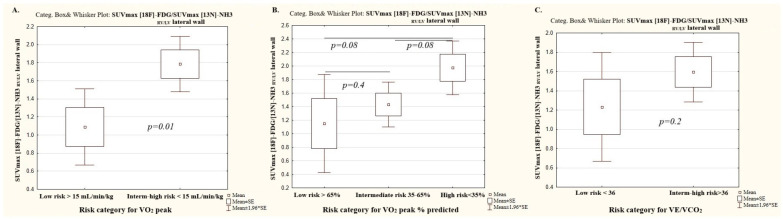
The ratio of metabolism to perfusion of the RV/LV myocardium depending on the risk categories of the main cardiorespiratory test parameters. (**A**) Peak oxygen consumption; (**B**) peak oxygen consumption % of predicted; (**C**) Minute ventilation per unit carbon dioxide production.

**Figure 3 diagnostics-15-02523-f003:**
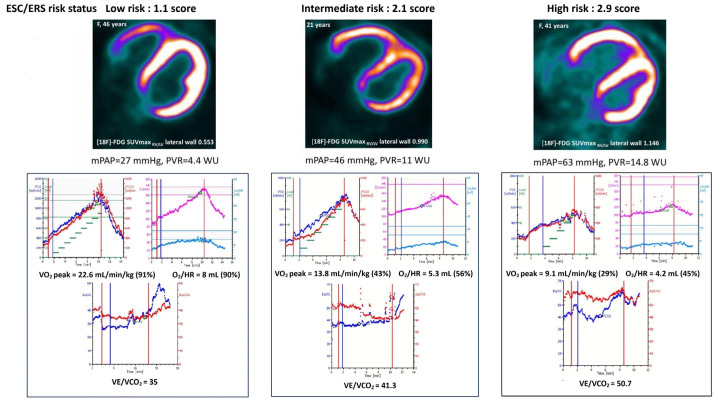
Examples of low-, intermediate- and high-risk IPAH patients, presented in terms of PET/CT SUVmax _RV/LV_ for [18F]-FDG, CPET data and hemodynamics.

**Table 1 diagnostics-15-02523-t001:** Characteristics of IPAH patients according to ESC/ERS 2022 risk status.

Parameters, *n* (%); M ± SD; Me [IQR 25; 75].	Entire Cohort, *n* = 34	Low Risk, *n* = 11	Intermediate Risk, *n* = 17	High Risk, *n* = 6	*p*-Value (*, **, ***)
Age (years)	33.9 ± 8.7	33.1 ± 7.8	34.4 ±10.2	34.5 ± 7.2	0.9 (0.7; 0.9; 0.7)
Male, n (%)	3 (8.8)	2 (18.1)	1 (5.8)	0 (0)	n/a
BMI (kg/m^2^)	25.3 ± 6.5	24.2 ± 4.0	26.2 ± 7.8	26.3 ± 7.3	0.7 (0.4; 0.9; 0.4)
Smoking, n (%)	6 (17.6)	3 (27)	2 (11.7)	1 (16.7)	n/a
Prevalent, n (%)	6 (17.6)	3 (27.2)	1 (5.8)	2 (33.3)	n/a
FC III-IV, n (%)	20 (58.8)	0	14 (82)	6 (100)	0.0002
6MWT, m	394.1 ±121.2	469.1 ± 71.6	365.7 ± 101.9	319.7 ± 178.2	0.01 (0.006; 0.4; 0.02)
**Laboratory parameters**					
NT-proBNP, pg/ml	767 [144; 2835]	114.0 [55.5; 210.0]	1016.0 [532.9; 2796.0]	4443.0 [3450.0; 5576.0]	0.00005 (0.006; 0.006;.000001)
CRP, mg/l	2.2 ± 1.9	1.2 ±0.6	2.2 ±1.6	4.2 ±3.1	0.006 (0.04; 0.06; 0.005)
eGFR, ml/min/1.73 m^2^	97.0 ± 28.3	100.7 ± 25.9	99.7 ± 25.5	82.7 ± 39.4	0.4 (0.9; 0.2; 0.2)
Hemoglobin, g/l	144.1 ± 17.0	136.4 ±19.4	148.2 ± 14.4	148.7 ± 15.7	0.1 (0.07; 0.9; 0.2)
**Right heart catheterization**					
mBP, mm Hg	86.2 ± 11.7	92.6 ± 13.4	81.8 ± 9.3	85.0 ± 9.3	0.047 (0.02; 0.4; 0.2)
mPAP, mm Hg	55.7 ± 19.8	40.8 ± 13.9	63.1 ± 18.5	65.8 ± 17.5	0.002 (0.002; 0.7; 0.004)
RAP, mm Hg	7.1 ± 5.4	3.8 ± 2.8	6.7 ± 4.4	14.8 ± 4.6	0.00005 (0.058; 0.001; 0.000009)
PCWP, mm Hg	8.4 ± 3.6	7.7 ± 4.5	8.2 ±3.3	10.2 ± 1.7	0.3 (0.7; 0.2; 0.2)
CI, l/min/m^2^	2.5 ± 0.8	3.3 ± 0.5	2.2 ± 0.4	1.8 ± 0.3	<0.000001 (0.000001; 0.05; 0.000004)
PVR, WU	11.3 [6.5; 15.5]	5.5 [3.9; 7.2]	12.5 [11.0; 16.7]	22.0 [14.8; 34.7]	0.0002 (0.001; 0.1; 0.00004)
PAC, ml/mm Hg	1.5 ± 1.0	2.3 ±0.8	1.3 ± 0.9	0.7 ±0.3	0.0004 (0.004; 0.1; 0.0002)
Sat O_2_, %	95.3 ± 2.8	96.3 ± 2.9	95.0 ± 2.7	94.2 ± 2.6	0.3 (0.2; 0.5; 0.1)
SvO_2_, %	62,4 ± 12,3	72.2 ± 5.8	59.9 ± 10.4	49.2 ± 12.0	0.00007 (0.001; 0.05; 0.00004)
**Cardiopulmonary exercise testing**					
Load, W	73.6 [50.0; 90.0]	90.0 [87.0; 102.5]	60.0 [50.0; 75.0]	50.0 [50.0; 70.0]	0.001 (0.001; 0.7; 0.03)
VO_2_ peak, mL/min/kg	13.4 [11.1; 18]	18.1 [16.9; 22.8]	11.7 [10.8; 14.2]	11.7 [10.8; 13.8]	0.0003 (0.0002; 0.9; 0.01)
VO_2_ peak predicted, %	54.4 ± 19.5	67.8 ± 19.2	47.3 ± 15.1	46.7 ± 18.6	0.008 (0.004; 0.9; 0.04)
%VO_2_/kg AT predicted, mL/kg/min	50.3 ± 17.8	61.8 ± 16.2	45.4 ± 13.3	39.8 ± 21.7	0.02 (0.01; 0.5; 0.04)
ΔVO_2_/ΔWR, mL/min/W	9.0 ± 2.4	10.2 ± 2.1	8.5 ± 2.4	7.8 ± 2.2	0.07 (0.06; 0.5; 0.04)
VO_2_/HR predicted, %	62.7 ± 19.0	72.7 ± 20.1	58.1 ± 17.9	55.3 ± 13.2	0.07 (0.05; 0.7; 0.07)
HR/Vkg, L/mL/kg	11.0 ± 2.8	9.1 ± 2.2	12.0 ± 2.7	12.2 ± 2.2	0.009 (0.006; 0.9; 0.01)
VE max predicted,%	60.2 ± 17.5	66.3 ± 20.1	59.4 ± 16.2	50.3 ± 12.1	0.1 (0.3; 0.2; 0.09)
Desaturation, %	2 [1; 4]	1.5 [1.0; 4.0]	2.0 [1.0; 4.0]	3.0 [2.0; 5.0]	0.7 (0.7; 0.6; 0.5)
ΔVD/VT, %	2.8 ± 6.8	6.3 ± 7.7	1.2 ± 6.3	1.2 ± 4.8	0.1 (0.07; 0.9; 0.1)
BR predicted, %	171.4 ± 48.4	146.4 ± 57.9	177.0 ± 39.9	201.0 ± 37.5	0.2 (0.2; 0.3; 0.1)
VE/VCO_2_	48.9 ± 13.1	42.6 ± 13.6	53.5 ± 12.4	49.0 ± 10.2	0.09 (0.04; 0.4; 0.3)
VE/VCO_2_ AT	45.2 ± 12.5	37.7 ± 8.4	50.9 ± 13.9	45.2 ± 8.0	0.03 (0.01; 0.4; 0.1)
PetCO_2_ rest	3.38 ± 0.7	3.73 ± 0.74	3.22 ± 0.73	3.16 ± 0.45	0.1 (0.09; 0.8; 0.1)
PetCO_2_ peak	3.06 ± 0.92	3.47 ± 0.92	2.8 ± 0.92	2.9 ± 0.7	0.2 (0.07; 0.6; 0.3)
∆ PetCO_2_	0.32 ± 0.49	0.25 ± 0.55	0.42 ± 0.41	0.17 ± 0.59	0.5 (0.4; 0.3; 0.7)
PetCO_2_ AT	3.29 ± 1.07	3.72 ± 1.29	2.97 ± 0.9	3.29 ± 0.7	0.3 (0.1; 0.5; 0.5)
**Cardiac MRI**					
RA short dimension, mm	49.6 ± 9.8	44.0 ± 4.9	52.4 ± 9.4	54.3 ± 13.7	0.03 (0.01; 0.7; 0.03)
RV EDV index, mL/m^2^	86.2 ± 22.6	79.7 ± 16.6	80.8 ± 18.7	112.1 ± 25.7	0.004 (0.8; 0.006; 0.005)
RV ESV index, mL/m^2^	57.1 ± 18.8	43.9 ± 12.3	56.4 ±10.5	85.2 ± 13.9	0.00001 (0.01; 0.00007; 0.000008)
RV wall thickness, mm	6.0 ± 1.8	5.0 ± 1.3	6.4 ±1.9	7.4 ± 1.6	0.02 (0.049; 0.2; 0.003 )
RV EF, %	36.5 ± 11.3	45.0 ±7.5	34.9 ± 8.8	23.2 ± 8.9	0.00005 (0.004; 0.01; 0.00005)
LA short dimension, mm	29.6 ± 5.4	33.1 ± 5.0	28.0 ± 4.3	26.2 ± 4.8	0.007 (0.01; 0.4; 0.01)
LV EDV index, mL/m^2^	58.9 ± 15.9	72.7 ± 9.2	55.3 ± 12.4	39.8 ± 7.6	0.000002 (0.0005; 0.01; 0.000001)
LV ESV index, mL/m^2^	23.3 ± 7.5	29.4 ± 6.2	20.8 ±5.4	17.0 ± 5.7	0.0002 (0.001; 0.2; 0.0009)
LV SV index, mL/m^2^	37.2 ± 14.6	48.1 ± 15.6	34.2 ± 8.9	22.6 ± 4.2	0.0003 (0.009; 0.007; 0.001)
LV EF, %	60.4 ± 6.3	60.8 ± 4.7	61.6 ± 6.4	57.0 ± 8.4	0.3 (0.7; 0.2; 0.2)
RV EDVi/LV EDVi	1.4 [1.03; 1.87]	1.0 [0.9; 1.2]	1.5 [1.2; 1.7]	2.7 [2.3; 4.1]	0.000001 (0.01; 0.0003; 0.000009)
RV ESVi/LV ESVi	2.37 [1.59; 3.56]	1.4 [1.3; 1.6]	2.7 [2.2; 3.3]	4.6 [4.4; 7.6]	0.000001 (0.00004; 0.001; 0.00004)
**PET-CT metabolism of the RV/LV**					
[18F]-FDG SUVmax _RV/LV_ lateral wall	0.866 ± 0.365	0.581 ± 0.393	0.941 ± 0.247	1.186 ± 0.293	0.002 (0.007; 0.07; 0.01)
**PET-CT perfusion of the RV/LV**					
[13N]-NH3 SUVmax _RV/LV_ lateral wall	0.817 ± 0.157	0.744 ± 0.153	0.867 ± 0.142	0.796 ± 0.184	0.1 (0.04; 0.3; 0.5)
**The ration of metabolism to perfusion of the RV/LV**					
SUVmax 18F-FDG/SUVmax [13N]-NH3 _RV/LV_ lateral wall	1.481 ± 0.799	0.786 ± 0.321	1.646 ± 0.646	2.311 ± 0.931	0.0002 (0.0006; 0.08; 0.0004)
**PAH therapy**					
Naïve patients, n (%)	28 (82.3)	3 (27.3)	1 (5.8)	2 (33.3)	n/a

Footnote: * difference between low and intermediate risk groups; ** difference between the intermediate and high risk groups; *** difference between low and high risk groups; AT, anaerobic threshold; BR, breathing reserve; BMI, body mass index; CI, cardiac index; CCB, calcium channel blockers; dO_2_/dW, the slope between the change in oxygen consumption relative to the change in work rate; EDVi, end–diastolic volume index; ESVi, end–systolic volume index; eGFR, estimated glomerular filtration rate; EF, ejection fraction; ESC, European Society of Cardiology; ERS, European Respiratory Society; ERA, endothelin receptor antagonist; FC, functional class; HR/Vkg, heart rate slope; IQR, interquartile range; LA, left atrial; LV, left ventricle; max, maximal; mBP, mean blood pressure; mPAP, mean pulmonary artery pressure; MRI, magnetic resonance imaging; n/a, not applicable; NT-proBNP, N-terminal pro-brain-type natriuretic peptide; PAC, pulmonary artery compliance; PCWP, pulmonary capillary wedge pressure; PET, positron emission tomography; Pet CO_2_, partial pressure of end-tidal carbon dioxide; PVR, pulmonary vascular resistance; RA, right atrial; RAP, right atrial pressure; RHC, right heart catheterization; RV, right ventricle; Sat O_2_, arterial oxygen saturation; SV, stroke volume; SUV, standardized uptake value; SvO_2_, mixed venous oxygen saturation; 6MWT, six minute walk test; [18F]-FDG, 18F-fluorodeoxyglucose; [13N]-NH3, ammonia; VE, ventilator equivalent; VE/VCO_2_, minute ventilation per unit carbon dioxide production; VO_2_/HR, oxygen pulse; VO_2_ peak, peak oxygen consumption; VO_2_/kgAT, peak oxygen consumption at anaerobic threshold; ΔVO_2_/ΔWR, the ratio between oxygen consumption and workload; ΔVD/VT, delta peak to rest of the ratio of the dead space volume to the tidal volume.

**Table 2 diagnostics-15-02523-t002:** Correlations between CPET parameters and hemodynamics.

Parameters	Mean PAP			Mean RAP			CI			PAC		
	r	t	** *p* **	r	t	** *p* **	r	t	** *p* **	r	t	** *p* **
%Load predicted	−0.57	−3.9	0.0004	−0.003	−0.02	0.9	0.44	2.8	0.009	0.48	3.2	0.003
VO_2_ peak	−0.45	−2.9	0.007	−0.17	−0.9	0.3	0.48	3.1	0.004	0.35	2.1	0.04
%VO_2_ peak predicted	−0.45	−2.9	0.007	−0.05	−0.3	0.7	0.52	3.4	0.002	0.49	3.2	0.003
VO_2_/HR	−0.37	−2.3	0.03	−0.07	−0.4	0.7	0.44	2.7	0.01	0.52	3.5	0.002
% VO_2_/HR predicted	−0.49	−3.2	0.003	−0.06	−0.3	0.7	0.38	2.3	0.02	0.47	3.1	0.004
∆VD/VT	0.14	0.8	0.4	0.04	0.2	0.8	0.12	0.6	0.5	0.10	0.5	0.5
%BR	0.14	0.6	0.5	0.16	0.7	0.5	−0.39	−1.8	0.08	−0.43	−2.1	0.05
PetCO_2_ rest	−0.38	−2.3	0.03	−0.01	−0.08	0.9	0.42	2.5	0.02	0.36	2.2	0.04
PetCO_2_ peak	−0.45	−2.8	0.008	0.003	0.01	0.9	0.34	2.0	0.05	0.38	2.3	0.03
∆Pet CO_2_	0.23	1.3	0.2	−0.04	−0.2	0.8	−0.007	−0.04	0.9	−0.15	−0.8	0.4
PetCO_2_ AT	−0.35	1.9	0.06	−0.004	−0.002	0.9	0.31	1.7	0.1	0.24	1.3	0.2
ΔVO_2_/ΔWR	−0.16	−0.8	0.4	−0.12	−0.7	0.5	0.45	2.7	0.01	0.27	1.5	0.1
HR/Vkg	0.52	3.3	0.002	0.14	0.8	0.4	−0.41	−2.5	0.02	−0.41	−2.5	0.02
VO_2_ AT	−0.41	−2.4	0.02	−0.36	−2.0	0.05	0.54	3.3	0.003	0.43	2.5	0.02
%VO_2_ AT predicted	−0.45	−2.6	0.01	−0.13	−0.7	0.5	0.56	3.5	0.002	0.53	3.2	0.004
VE/VCO_2_	0.37	2.3	0.03	0.11	0.6	0.5	−0.37	−2.3	0.03	−0.28	−1.7	0.1
VE/VCO_2_ AT	0.43	2.4	0.02	0.03	0.1	0.9	−0.44	−2.5	0.02	−0.31	−1.7	0.1

Footnotes: AT, anaerobic threshold; BR, breathing reserve; CI, cardiac index; dO_2_/dW, the slope between the change in oxygen consumption relative to the change in work rate; HR/Vkg, heart rate slope; PAC, pulmonary artery compliance; PAP, pulmonary artery pressure; PetCO_2_, partial pressure of end-tidal carbon dioxide; RAP, right atrial pressure; PVR, pulmonary vascular resistance; VO_2_/HR, oxygen pulse; VO_2_ peak, peak oxygen consumption; VO_2_/kgAT, peak oxygen consumption at anaerobic threshold; VE/VCO_2_, minute ventilation per unit carbon dioxide production; ΔVO_2_/ΔWR, relationship between oxygen consumption and workload; ΔVD/VT, delta peak to rest of the ratio of the dead space volume to the tidal volume; VO_2_/HR, oxygen pulse.

**Table 3 diagnostics-15-02523-t003:** Correlations between CPET parameters and cardiac MRI.

Parameters	RV ESV Index			LV SV Index			RV EF			RV ESV Index/LV ESV Index		
	r	t	** *p* **	r	t	** *p* **	r	t	** *p* **	r	t	** *p* **
%Load predicted	−0.38	−2.25	0.03	0.44	2.6	0.01	0.3	1.8	0.08	−0.58	−3.9	0.0005
VO_2_ peak	−0.29	−1.67	0.1	0.54	3.5	0.001	0.3	1.7	0.1	−0.66	−4.8	0.00004
%VO_2_ peak predicted	−0.37	−2.19	0.036	0.49	3.1	0.004	0.3	1.5	0.1	−0.60	−4.1	0.0003
VO_2_/HR	−0.26	−1.46	0.1	0.43	2.6	0.01	0.17	0.9	0.3	−0.42	−2.5	0.02
%VO_2_/HR predicted	−0.27	−1.5	0.1	0.46	2.8	0.008	0.29	1.7	0.1	−0.46	−2.8	0.008
∆VD/VT	0.1	0.6	0.5	0.1	0.5	0.6	0.06	0.3	0.7	−0.02	−0.1	0.9
%BR	0.2	1.0	0.3	−0.02	−0.1	0.9	0.03	0.1	0.8	0.3	1.3	0.2
PetCO_2_ rest	−0.39	−2.3	0.03	0.3	1.8	0.07	0.2	1.1	0.2	−0.53	−3.3	0.002
PetCO_2_ peak	−0.35	−2.02	0.05	0.42	2.5	0.02	0.2	1.3	0.2	−0.52	−3.3	0.002
∆Pet CO_2_	0.09	0.5	0.6	−0.26	−1.4	0.1	−0.09	−0.5	0.6	0.22	1.2	0.2
PetCO_2_ AT	−0.36	−1.8	0.07	0.30	1.5	0.1	0.18	0.9	0.4	−0.43	−2.4	0.02
ΔVO_2_/ΔWR	−0.21	−1.1	0.3	0.44	2.6	0.01	0.2	1.1	0.3	−0.44	−2.6	0.01
HR/Vkg	0.22	1.2	0.2	−0.55	−3.5	0.002	−0.27	−1.5	0.1	0.54	3.4	0.002
VO_2_ AT	−0.44	−2.5	0.02	0.56	3.4	0.002	0.2	1.2	0.2	−0.69	−4.8	0.00005
%VO_2_ AT predicted	−0.47	−2.6	0.01	0.49	2.8	0.01	0.2	1.3	0.2	−0.63	−3.9	0.0005
VE/VCO_2_	0.30	1.7	0.09	−0.42	−2.5	0.016	−0.19	−1.1	0.3	0.51	3.3	0.002
VE/VCO_2_ AT	0.47	2.6	0.01	−0.39	−2.1	0.048	−0.33	−1.7	0.09	0.57	3.4	0.002

Footnotes: AT, anaerobic threshold; BR, breathing reserve; ESV, end systolic volume; EF, ejection fraction; HR/Vkg, heart rate slope; LV, left ventricle; PetCO_2_, partial pressure of end-tidal carbon dioxide; RV, right ventricle; SV, stroke volume; VO_2_/HR, oxygen pulse; VO_2_ peak, peak oxygen consumption; VO_2_/kgAT, peak oxygen consumption at anaerobic threshold; VE/VCO_2_, minute ventilation per unit carbon dioxide production; ΔVO_2_/ΔWR, relationship between oxygen consumption and workload; ΔVD/VT, delta peak to the rest of the ratio of the dead space volume to the tidal volume; VO_2_/HR, oxygen pulse.

**Table 4 diagnostics-15-02523-t004:** Correlations between [18F]-FDG and [13N]-NH3 uptake of the RV/LV lateral wall and CPET parameters.

Parameters	[18F]-FDG SUVmax _RV/LV_ Lateral Wall			[13N]-NH3 SUVmax _RV/LV_ Lateral Wall			[18F]-FDG SUVmax/[13N]-NH3 SUVmax _RV/LV_ Lateral Wall		
	r	t	** *p* **	r	t	** *p* **	r	t	** *p* **
%Load predicted	−0.54	−3.5	0.002	−0.58	−4.1	0.0003	−0.45	−2.7	0.009
VO_2_ peak	−0.59	−4.1	0.0003	−0.46	−2.9	0.006	−0.54	−3.5	0.01
%VO_2_ peak predicted	−0.46	−2.9	0.007	−0.51	−3.3	0.002	−0.46	−2.8	0.008
VO_2_/HR	−0.22	1.2	0.2	−0.39	−2.4	0.02	−0.35	−2.04	0.05
% VO_2_/HR predicted	−0.29	−1.7	0.09	−0.49	−3.1	0.003	−0.25	−1.5	0.1
∆VD/VT	0.16	0.89	0.4	0.37	2.2	0.03	−0.28	−1.6	0.1
%BR	0.03	0.1	0.9	0.08	0.4	0.7	0.06	0.2	0.8
PetCO_2_ rest	−0.46	−2.8	0.009	−0.60	−4.2	0.0002	−0.26	−1.4	0.2
PetCO_2_ peak	−0.38	−2.2	0.03	−0.44	−2.7	0.009	−0.26	−1.5	0.2
∆Pet CO_2_	0.08	0.4	0.6	0.04	0.2	0.8	0.1	0.7	0.5
PetCO_2_ AT	−0.44	−2.4	0.02	−0.59	−3.7	0.0009	−0.25	−1.3	0.2
ΔVO_2_/ΔWR	−0.28	−0.6	0.1	−0.27	−1.5	0.1	−0.35	−1.9	0.057
HR/Vkg	0.45	2.7	0.01	0.53	3.4	0.002	0.42	2.5	0.02
VO_2_ AT	−0.56	−3.4	0.002	−0.46	−2.7	0.01	−0.47	−2.70	0.01
%VO_2_ AT predicted	−0.52	−3.1	0.005	−0.54	−3.3	0.003	−0.44	−2.4	0.02
VE/VCO_2_	0.37	2.2	0.03	0.36	2.2	0.04	0.2	1.1	0.3
VE/VCO_2_ AT	0.44	2.5	0.02	0.62	4.02	0.0004	0.3	1.7	0.1

Footnotes: AT, anaerobic threshold; BR, breathing reserve; EF, ejection fraction; HR/Vkg, heart rate slope; LV, left ventricle; PET, positron emission tomography; PetCO_2_, partial pressure of end-tidal carbon dioxide; RV, right ventricle; SV, stroke volume; SUV, standardized uptake value; VO_2_/HR, oxygen pulse; VO_2_ peak, peak oxygen consumption; VO_2_/kgAT, peak oxygen consumption at anaerobic threshold; VE/VCO_2_, minute ventilation per unit carbon dioxide production; ΔVO_2_/ΔWR, relationship between oxygen consumption and workload; ΔVD/VT, delta peak to the rest of the ratio of the dead space volume to the tidal volume; VO_2_/HR, oxygen pulse; [18F]-FDG, 18F-fluorodeoxyglucose; [13N]-NH3, ammonia.

**Table 5 diagnostics-15-02523-t005:** [18F]-FDG and [13N]-ammonia uptake by RV myocardium depending on the risk categories of the main cardiopulmonary test determinants.

	Peak Oxygen Consumption, mL/min/kg			
Parameters, *n* (%); m ± SD	Low Risk (*n* = 14)	Intermediate/High Risk (*n* = 20)		*p* Value
Risk Category	>15	<15		
[18F]-FDG SUVmax _RV/LV_ lateral wall	0.59 ± 0.28	1.08 ± 0.26		0.00002
[13N]-NH3 SUVmax _RV/LV_ lateral wall	0.72 ± 0.10	0.89 ± 0.15		0.0005
SUVmax 18F-FDG/SUVmax [13N]-NH3 _RV/LV_ lateral wall	1.08 ± 0.8	1.78 ± 0.6		0.01
	**Peak oxygen consumption,** **%, Predicted**			
**Risk category**	**Low risk** (*n* = 8)	**Intermediate** (*n* = 17)	**High risk** (*n* = 9)	***p* value,** all groups (*;**;***)
	**>65**	**35–65**	**<35**	
[18F]-FDG SUVmax _RV/LV_ lateral wall	0.6 ± 0.3	0.9 ± 0.3	1.2 ± 0.3	0.01 (0.1; 0.04; 0.005)
[13N]-NH3 SUVmax _RV/LV_ lateral wall	0.7 ± 0.07	0.8 ± 0.1	1.0 ± 0.2	0.003 (0.04; 0.04; 0.001)
SUVmax 18F-FDG/SUVmax [13N]-NH3 _RV/LV_ lateral wall	1.2 ± 1.0	1.4 ± 0.7	2.0 ± 0.5	0.1 (0.4; 0.08; 0.08)
	**VE/VCO_2_, the ratio of minute ventilation to carbon dioxide production**			
**Risk category**	**Low risk** (*n* = 10)	**Intermediate/high risk** (*n* = 24)		*p* value
	**<36**	**>36**		
[18F]-FDG SUVmax _RV/LV_ lateral wall	0.6 ± 0.3	0.9 ± 0.3		0.007
[13N]-NH3 SUVmax _RV/LV_ lateral wall	0.7 ± 0.09	0.8 ± 0.16		0.03
SUVmax 18F-FDG/SUVmax [13N]-NH3 _RV/LV_ lateral wall	1.2 ± 0.9	1.6 ± 0.7		0.2

Footnotes: * difference between low and intermediate risk groups; ** difference between intermediate and high risk group; *** difference between low and high risk groups; [18F]-FDG, 18F-fluorodeoxyglucose; [13N]-NH3, ammonia; VE/VCO_2_, minute ventilation per unit carbon dioxide production; VO_2_ peak, peak oxygen consumption; VO_2_/kgAT, peak oxygen consumption at anaerobic threshold; LV, left ventricle; max, maximal; RV, right ventricle; SUV, standardized uptake value.

**Table 6 diagnostics-15-02523-t006:** Linear regression analysis of PET/CT and cardiac MRI parameters associated with VO_2_ peak and VE/VCO_2_.

Factors	OR	95% CI	*p* Value
Univariate Regression Analysis for VO_2_ Peak			
[18F]-FDG SUVmax _RV/LV_ lateral wall	−7.97	−11.9–−3.97	0.0003
[13N]-NH3 SUVmax _RV/LV_ lateral wall	−0.02	−0.17–0.12	0.7
SUVmax 18F-FDG/SUVmax [13N]-NH3 _RV/LV_ lateral wall	−0.06	−0.117–−0.013	0.015
RV ESV index, mL/m^2^	−1.22	−2.46–0.028	0.055
RV ESV index/LV ESV index	−0.28	−2.15–−0.41	0.005
RV EF, %	0.76	0.016–1.503	0.045
LV SV index, mL/m^2^	1.17	0.23–2.1	0.016
**Multivariate regression analysis for VO_2_ peak**			
[18F]-FDG SUVmax _RV/LV_ lateral wall	−6.4	−11.5–−1.25	0.016
RV ESV index/LV ESV index	−0.55	−1.6–0.5	0.3
RV EF, %	0.002	−0.16–0.16	0.9
**Univariate regression analysis for VE/VCO_2_**			
[18F]-FDG SUVmax _RV/LV_ lateral wall	14.0	2.2–25.8	0.02
[13N]-NH3 SUVmax _RV/LV_ lateral wall	38.2	11.7–64.7	0.006
SUVmax 18F-FDG/SUVmax [13N]-NH3 _RV/LV_ lateral wall	1.7	−4.2–7.6	0.5
RV ESV index, mL/m^2^	0.15	−0.009–0.4	0.2
RV ESV index/LV ESV index	2.1	−0.37–4.5	0.09
RV EF, %	−0.16	−0.58–0.26	0.4
LV SV index, mL/m^2^	−0.27	−0.59–0.04	0.08
**Multivariate regression analysis for VE/VCO_2_**			
[18F]-FDG SUVmax _RV/LV_ lateral wall	2.9	−14.4–20.4	0.7
[13N]-NH3 SUVmax _RV/LV_ lateral wall	30.5	−4.4–65.5	0.08
RV ESV index/LV ESV index	0.9	−1.8–3.8	0.5

Footnotes: ESV, end systolic volume; EF, ejection fraction; LV, left ventricle; RV, right ventricle; SV, stroke volume; SUV, standardized uptake value; VE/VCO_2_, minute ventilation per unit carbon dioxide production; VO_2_ peak, peak oxygen consumption; [18F]-FDG, 18F-fluorodeoxyglucose; [13N]-NH3, ammonia.

## Data Availability

The data supporting the conclusions of this study are available from the corresponding author upon reasonable request.

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
