# Peer review of "Right Ventricular Myocardial Metabolism and Cardiorespiratory Testing in Patients with Idiopathic Pulmonary Arterial Hypertension"

_diagnostics, 2025, doi:10.3390/diagnostics15192523_

Round 1

Reviewer 1 Report

Comments and Suggestions for Authors

This study is the first to explore the relationship between right ventricular metabolism and CPET parameters in IPAH patients, demonstrating certain clinical significance. However, the following issues remain:

  1. According to 2022 ESC/ERS Guidelines for the diagnosis and treatment of pulmonary hypertension, the diagnostic criteria for PAH is mPAP>20mmHg,PAWP≤15mmHg and PVR>2 WU.It is recommended that PAH patients be included in the latest international guidelines.
  2. This study included a small number of patients (only 34 cases) and did not undergo prospective validation, resulting in limited reference value. It is recommended that future studies include a larger patient population and undergo validation.Additionally, the study population consisted of Caucasians from a single center, which leads to limited statistical power and insufficient sample size and representativeness.
  3. There is a mixture of incident patients, previously treated patients, and those positive for calcium channel blockers. The potential impact of drugs on metabolism-perfusion coupling has not been adjusted for.Some patients received different PAH targeted therapies (such as calcium channel blockers, triple combination therapy, etc.), but the potential impact of treatment regimens on right ventricular metabolism, perfusion, and CPET parameters was not analyzed, which may introduce confounding biases. Meanwhile, the influence of baseline characteristics such as age and gender on the results was not explored.
  4. In Table 3, the correlation between RV EF and CPET indicators is relatively weak.
  5. The specificimplementation details ofthe insulin-normal glucose clamp technique (e.g. insulin dose, glucose stabilization time) are described briefly, May affect the stability of [ 18F ] - FDG uptake.
  6. The results of " no significant difference between the medium and high risk groups " were not analyzed in depth, but only due to the small sample size. Did not consider whether there was a " plateau " of disease progression or insufficient sensitivity of the markers.
  7. It is recommended to supplement the specific number of patients for each examination in Table 1 to clarify whether all parameters are complete for each sample.
  8. n the Discussion section, the authors should further elaborate on how the findings can guide clinical practice, such as their application in risk stratification or treatment monitoring.

Author Response

We thank the Reviewer for the careful evaluation of our manuscript and valuable comments.

We have revised the manuscript according to all your comments and suggestions.

Please, find below our point-by-point responses.

Comment 1: According to 2022 ESC/ERS Guidelines for the diagnosis and treatment of pulmonary hypertension, the diagnostic criteria for PAH is mPAP>20mmHg,PAWP≤15mmHg and PVR>2 WU. It is recommended that PAH patients be included in the latest international guidelines.

Response: Patients were enrolled from February 2020 to October 2024. According to the national guidelines 2020 for the management of pulmonary hypertension the following hemodynamic criteria for PAH were used in the study in that period: mean pulmonary artery preassure (PAP) ≥25 mmHg, pulmonary capillary wedge pressure (PCWP) < 15 mmHg and pulmonary vascular resistance (PVR) ≥3 Wood units [Reference 13: S.N. Avdeev, O.L. Barbarash, A.E. Bautin et al., 2020 Clinical practice guidelines for Pulmonary hypertension, including chronic thromboembolic pulmonary hypertension, Russ J Cardiol. 26 (2021) 4683. https://doi.org/10.15829/1560-4071-2021-4683]. The new definition with the mean PAP>20 mmHg and PVR>2 WU has been implemented into the clinical practice with the official publication of the national PH recommendations on October, 2024 [Reference 15: S.N. Avdeev, O.L. Barbarash, Z.S. Valieva et al. 2024 Clinical practice guidelines for Pulmonary hypertension, including chronic thromboembolic pulmonary hypertension, Russ J Cardiol. 29 (2024) 6161. (In Russ.) https://doi.org/10.15829/1560-4071-2024-6161.].

Comment 2: This study included a small number of patients (only 34 cases) and did not undergo prospective validation, resulting in limited reference value. It is recommended that future studies include a larger patient population and undergo validation. Additionally, the study population consisted of Caucasians from a single center, which leads to limited statistical power and insufficient sample size and representativeness.

Response: We agree with the Reviewer that the larger patient population and long-term follow-up improve statistical power and the results representativeness. We continue PET/CT study and prospective follow-up of all patients.

Limited access to the PET/CT equipment in other PH centers of the country and high cost of the study are the main limitations for the multicenter study. Caucasians is the predominant race in our region. At present, we do not have other races in our PAH registry.

We added to the “Limitation” section:

The sample size was small (n=34), with the only 6 patients in the high-risk group, potentially affecting statistical power (e.g., the lack of significant differences between intermediate- and high-risk groups).

The prognostic value of CPET, as well as PET/CT data, could not be assessed due to an insufficient observation period.

Comment 3: There is a mixture of incident patients, previously treated patients, and those positive for calcium channel blockers. The potential impact of drugs on metabolism-perfusion coupling has not been adjusted for. Some patients received different PAH targeted therapies (such as calcium channel blockers, triple combination therapy, etc.), but the potential impact of treatment regimens on right ventricular metabolism, perfusion, and CPET parameters was not analyzed, which may introduce confounding biases. Meanwhile, the influence of baseline characteristics such as age and gender on the results was not explored.

Response:  

  1. Indeed, the low number of prevalent patients did not permit to analyze the influence of calcium channel blockers (n=3) or PAH therapies (n=3) on the right ventricular metabolism and perfusion.

We could point out, that the patient selection for the study was meticulous with exclusion of left heart disease, respiratory comorbidities, diabetes mellitus and obesity with BMI>30 kg/m2 in order to exclude the possibility of coronary atherosclerosis disease or pulmonary hypertension, associated with the persevered left ventricle ejection fraction with the stiff left heart. We performed right heart catheterization and all diagnostic procedures in prevalent patients at the time of PET/CT studies.

Moreover, it is difficult to assess the true influence of PAH-therapy on metabolism and perfusion, as these processes are tightly connected with the value of RV pressure overload in IPAH patients.

We added to the “Results”section: “No significant differences were registered between prevalent (n=6) and incident patients in [18F]-FDG SUVmax RV/LV lateral wall (0.924±0.476 vs 0.825±0.348, p=0.5), [13N]-NH3 SUVmax RV/LV lateral wall (0.825±0.176 vs 0.815±0.157, p=0.8) and SUVmax 18F-FDG/SUVmax [13N]-NH3 RV/LV lateral wall (1.699±0.967 vs 1.381±0.763, p=0.4). Similar values of VO2 peak (12 [11.1; 18] vs 14.5 [11.1; 17.9] ml/min/kg, p=0.9), VO2 peak Predicted (54.2±31 vs 54.5±16.9%, p=0.9) and VE/VCO2 (44.6±9.6 vs 49.7±13.7, p=0.4) were observed in prevalent and incident patients”

  1. Three men only were in the study, which was not enough for the gender differences analysis.
  2. The mean age of investigated population was rather young 33.9 ± 8.7 years with normal distribution.

We added to the “Results” section: “No significant correlations were registered between age and hemodynamics, RV ESV index, RV EF and LV SV index, derived with cardiac MRI, PET/CT and CPET parameters (Supplements, Table 1.1.)”.

We added in Supplements: “Table 1.1. (Supplements) Correlations between age and the main hemodynamics, cardiac MRI, PET/CT and CPET indices in the entire cohort”.

Parameter

n

Spearman, ρ

t

p-value

Right heart catheterization

mPAP, mmHg

34

-0,008655

-0,048962

0,961254

PCWP, mmHg

34

0,315297

1,879457

0,069315

RAP, mmHg

34

0,055424

0,314006

0,755555

CI, l/min/m2

34

-0,032308

-0,182854

0,856066

PVR, WU

34

-0,022437

-0,126953

0,899772

PAC, mmHg/ml

34

0,077930

0,442184

0,661330

Cardiac MRI

RV ESV index, ml/m2

34

0,176043

1,011650

0,319294

RV EF, %

34

-0,271689

-1,59698

0,120100

LV SV index, ml/m2

34

-0,184003

-1,05896

0,297546

PET/CT

[18F]-FDG SUVmax RV/LV lateral wall

34

0,050364

0,285262

0,777282

[13N]-NH3 SUVmax RV/LV lateral wall

34

-0,263606

-1,54585

0,131973

SUVmax 18F-FDG/SUVmax [13N]-NH3 RV/LV lateral wall

34

0,172522

0,990790

0,329221

Cardiopulmonary exercise testing

VO2 peak, ml/min/kg

34

-0,051987

-0,294481

0,770293

VO2 peak Predicted, %

34

-0,178088

-1,02378

0,313616

V'O2/HR Predicted, %

34

0,090429

0,513648

0,611027

VE/VCO2

34

-0,009495

-0,05371

0,957498

Footnote: CI, cardiac index; CPET, cardiopulmonary exercise test; ESVi, end-systolic volume index; EF, ejection fraction; LV, left ventricle; max, maximal; mPAP, mean pulmonary artery pressure; MRI - magnetic resonance imaging; PAC, pulmonary artery compliance; PCWP, pulmonary capillary wedge pressure; PET, positron emission tomography; PVR, pulmonary vascular resistance; RAP, right atrial pressure; RV, right ventricle; SV, stroke volume: SUV, standardized uptake value; [18F]-FDG, 18F-Fluorodeoxyglucose; [13N]-NH3, ammonia; VE/VCO2, the ratio of minute ventilation to carbon dioxide production; VO2/HR, oxygen pulse; VO2 peak, peak oxygen consumption; VO2/HR, oxygen pulse.

Comment 4: In Table 3, the correlation between RV EF and CPET indicators is relatively weak.

Response: RV EF is one of prognostic parameters according to the ESC/ERS 2022 risk stratification scale. Thus, we presented data on correlations between RV EF and CPET parameters. We mentioned in “Discussion” section: “We did not find any significant correlations between CPET parameters and RV ejection fraction, which might be due to the small patient population and a wide range of intermediate-risk limits for the RV ejection fraction”. Indeed, in clinical practice we observe a wide range of RV EF in low-risk and intermediate-risk patients. This observation is in accordance with other publications [S. Alabed, P. Garg, F. Alandejani, et al. Establishing minimally important differences for cardiac MRI end-points in pulmonary arterial hypertension, Eur Respir J. (2023) 62: 2202225. https://doi.org/10.1183/13993003.02225-2022; ; https://doi.org/10.15829/15604071-2023-5540].

We added mentioned publication to the Reference.

Comment 5: The specific implementation details of the insulin-normal glucose clamp technique (e.g. insulin dose, glucose stabilization time) are described briefly, May affect the stability of [18F] - FDG uptake.

Response: We use hyperinsulinemic euglycemic clamp protocol to achieve the high quality of FDG PET image in all patients. This approach is especially relevant in patients with diabetes mellitus when glucose uptake is impaired.

We added to the “2.2. PET/CT Protocol” section:

“Cardiac PET/CT («Discovery 710», GE Healthcare, USA) with [18F]-FDG and [13N]-ammonia was performed in all patients in two separate days. Myocardial glucose metabolism was assessed by [18F]-FDG PET/CT. Patients fasted at least 6 hours before the procedure [18F]-FDG PET/CT. The hyperinsulinemic euglycemic clamp was performed as described in the publication by R. A. De Fronzo (1979) [16]. А polyethylene cannula was inserted into cubital vein to infuse insulin and 10% glucose solution through a three-way stopcock. Insulin was administrated at a constant rate of 40 mIU/min/m². The rate of intravenous glucose infusion was adjusted manually. Glucose level in blood was checked every 5 minutes. Stable euglycemia was considered to be achieved if three consecutive blood glucose measurements differed from each other within ±5%. Stable euglycemia was usually achieved 1 hour after the start of insulin and glucose infusion. A standard dose of 5 MBq/kg (<550 MBq) [18F]-FDG was administered intravenously, when stable glycemia was established (optimal glycemia was 5 mmol/L). Cardiac PET/CT scans were acquired in 40 minutes after [18F]-FDG intravenous administration in a static mode. Low dose CT scan was performed immediately prior to PET for attenuation correction. PET perfusion scanning was performed at rest, 5 minutes after intravenous administration of 10 MBq/kg [13N]-ammonia in a static mode immediately after low dose CT transmission.”.

Comment 6: The results of "no significant difference between the medium and high risk groups" were not analyzed in depth, but only due to the small sample size. Did not consider whether there was a "plateau" of disease progression or insufficient sensitivity of the markers.

Response: We presented p-value for all risk groups separately in Table 1 (Footnotes: * difference between low and intermediate risk groups; ** difference between intermediate and high-risk group; *** difference between low and high risk groups).

We did not analyze the course of the disease, so it is impossible to explain the lack of differences between patients with intermediate and high risk as a "plateau" of disease progression. In the future, with the accumulation of a sufficient observation period, we will evaluate the course of the disease from the standpoint of PET/CT, CRT indicators and their prognostic value.

Comment 7: It is recommended to supplement the specific number of patients for each examination in Table 1 to clarify whether all parameters are complete for each sample.

Response: There were 37 IPAH patients with PET/CT study. CPET data were not available in 3 patients and MRI in 1 patient. We enrolled 34 patients, who had the full list of examinations for the study.

We pointed in Table 1 the number of patients for each risk group, as well as in the other Tables and on Figures with correlations the number of cases is evident for each investigated parameter.

Comment 8: in the Discussion section, the authors should further elaborate on how the findings can guide clinical practice, such as their application in risk stratification or treatment monitoring.

Response: We were extremely cautious with the elaboration of our findings in PET/CT on the disease course, follow-up and prognostic stratification, as we do not have enough follow-up period and outcomes. All patients are alive and only several patients have repeat examinations including invasive hemodynamics.

We changed a part of the “Discussion” section:

“PET/CT studies have not been implemented into clinical practice for PAH patients yet, due to unclear practical significance. Our observation needs further data accumulation and analysis in terms of interconnection between RV metabolism and perfusion and hemodynamics, cardiac remodeling and outcome in IPAH patients. No specific drugs that target RV myocardium in PAH are currently exist. The approved PAH-specific medicines realize positive remodeling and metabolic effect on the RV through mPAP decrease, which was confirmed by R. Kazimierczyk et al. (2023). RV metabolism molecular imaging using PET/CT might be essential for the new treatment goals formation and the new drugs creation.

We clearly demonstrated significant correlations between CPET parameters and hemodynamics, cardiac MRI, consistent with the previous studies [10, 29, 30]. The established interconnection between CPET parameters and the RV metabolism are of a paramount importance, as the molecular changes preclude clinical manifestation and tightly connected with hemodynamics in IPAH patients [22, 25, 27]. Initial strategy and the first 6 months of IPAH patient management might determine the course of the disease [31]. We suggested that CPET might be an excellent non-invasive diagnostic tool for the low-risk status confirmation in young IPAH patients without comorbidities”

We have modified the “Conclusion” section:

Right ventricular [18F]-FDG myocardial uptake was an independent predictor of peak oxygen consumption in IPAH patients without comorbidities.

CPET has proven to be a reliable non-invasive tool for differentiation between IPAH patients without comorbidities with low-risk and those with intermediate-high risk status.

Accumulation of data on right ventricular myocardial metabolism in comparison with traditional invasive and non-invasive methods for PAH severity assessment requires prospective validation in a larger cohort of patients with IPAH.

Reviewer 2 Report

Comments and Suggestions for Authors

The authors report on cardiopulmonary exercise testing (CPET) and right ventricular (RV) perfusion and metabolism using positron emission tomography PET/CT with [13N]-ammonia and 18F]-FDG in 34 patients with idiopathic pulmonary arterial hypertension (IPAH) Significant negative correlations were observed between SUVmax RV/LV metabolism, SUVmax RV/LV perfusion and oxygen consumption, oxygen pulse and positive correlation with the ratio of minute ventilation to carbon dioxide production, all different between low vs intermediate or high risk status. Relationships between CPET and invasive hemodynamics, RV remodeling (cardiac magnetic resonance, CMR) were also found. The authors conclude that, accordingly, CPET is a reliable noninvasive diagnostic tool for distinguishing low-risk IPAH patients from intermediate-high risk population.

This is interesting and well reported, though with too much detail and over-abundance of correlations.

  1. The main problem is that there are many more correlations than patients. This is an unsolvable statistical multiplicity problem. A solution would be in a uni/multivariable analysis to identify independent RV function/metabolism predictors of aerobic exercise capacity. This could be done using simple models integrating RV perfusion or metabolism indices and CMR-derived EF and either VO2max, VE/VCO2 or 6min walk distance. The rest is less relevant as it is known that more severely ill patients have an impaired exercise capacity
  2. Another problem is the abundance of less relevant measurements such as lung function tests, which make the reading difficult. It is preferable to report on the measurements used to test your hypothesis. IPAH patients are known to have normal or near-normal lung function. In general less relevant measurements can be simply summarized in the text.
  3. I have no objection to the use of national PH guidelines, but the references list should be revised and updated to integrate higher quality papers and be more tightly related to the study.
  4. The English style is globally acceptable but inaccuracies need revision.
  5. Please define all abbreviations
  6. Please comment more clearly on the added value of this study compared to your previous publications
Comments on the Quality of English Language

Minor revision needed

Author Response

We thank the Reviewer for the careful evaluation of our manuscript and valuable comments.

We have revised the manuscript according to all your comments and suggestions.

Please, find below our point-by-point responses.

Comment 1: The main problem is that there are many more correlations than patients. This is an unsolvable statistical multiplicity problem. A solution would be in a uni/multivariable analysis to identify independent RV function/metabolism predictors of aerobic exercise capacity. This could be done using simple models integrating RV perfusion or metabolism indices and CMR-derived EF and either VO2max, VE/VCO2 or 6min walk distance. The rest is less relevant as it is known that more severely ill patients have an impaired exercise capacity.

Response: We shortened and simplified the description of the multiple correlations between CPET and hemodynamics, MRI and PET/CT in the “Results” section. We have omitted well-known correlations between CPET and hemodynamics (Table 2), CPET and cardiac MRI (Table 3). We would like to highlight the holistic picture of close correlations between CPET parameters, invasive hemodynamics and RV remodeling even in a small population of IPAH patients (n=34) to increase the value of CPET for the wide physicians audience.

We omitted Table 4 with correlations between CPET and PET/CT parameters in the Supplements section and presented Figure 1

Figure 1. Analysis of PET/CT and MRI parameters associated with the aerobic exercise capacity: A. [18F]-FDG SUVmax RV/LV lateral wall; B. [13N]-NH3 SUVmax RV/LV lateral wall; C. SUVmax 18F-FDG/SUVmax [13N]-NH3 RV/LV lateral wall; D. RV ESV index; E. LV SV index; F. RV EF; G. RV ESV index/LV ESV index.

We added paragraph in the “Results” section and Table 6:

3.11. PET/CT and MRI predictores for the peak oxygen consumption and minute ventilation per unit carbon dioxide production

Peak O2 consumption was significantly associated with myocardial RV uptake with [18F]-FDG, the ratio of metabolism to perfusion of RV, RV ejection fraction, the ratio of indices RV ESV/LV ESV and LV stroke volume index according to the univariate regression analysis. Myocardial RV uptake with [18F]-FDG was the independent predictor of peak O2 consumption according to the multivariate linear regression analysis (OR-6.5, 95% CI (-11.5-1.25), p=0.016). Minute ventilation per unit carbon dioxide production was significantly associated with the RV metabolism and perfusion according to the univariate regression analysis, but no independent predictor was revealed determined with miltivriate analysis (Table 6).

Table 6. Linear regression analysis of PET/CT and cardiac MRI parameters associated with VO2 peak and VE/VCO2

Factors

OR

95% CI

p value

Univariate regression analysis for VO2 peak

[18F]-FDG SUVmax RV/LV lateral wall

-7.97

-11.9--3.97

0.0003

[13N]-NH3 SUVmax RV/LV lateral wall

-0.02

-0.17-0.12

0.7

SUVmax 18F-FDG/SUVmax [13N]-NH3 RV/LV lateral wall

-0.06

-0.117- -0.013

0.015

RV ESV index, ml/m2

-1.22

-2.46-0.028

0.055

RV ESV index/LV ESV index

-0.28

-2.15- -0.41

0.005

RV EF, %

0.76

0.016-1.503

0.045

LV SV index, ml/m2

1.17

0.23-2.1

0.016

Multivariate regression analysis for VO2 peak

[18F]-FDG SUVmax RV/LV lateral wall

-6.4

-11.5- -1.25

0.016

RV ESV index/LV ESV index

-0.55

-1.6-0.5

0.3

RV EF, %

0.002

-0.16-0.16

0.9

Univariate regression analysis for VE/VCO2

[18F]-FDG SUVmax RV/LV lateral wall

14.0

2.2-25.8

0.02

[13N]-NH3 SUVmax RV/LV lateral wall

38.2

11.7-64.7

0.006

SUVmax 18F-FDG/SUVmax [13N]-NH3 RV/LV lateral wall

1.7

-4.2-7.6

0.5

RV ESV index, ml/m2

0.15

-0.009-0.4

0.2

RV ESV index/LV ESV index

2.1

-0.37-4.5

0.09

RV EF, %

-0.16

-0.58-0.26

0.4

LV SV index, ml/m2

-0.27

-0.59-0.04

0.08

Multivariate regression analysis for VE/VCO2

[18F]-FDG SUVmax RV/LV lateral wall

2.9

-14.4-20.4

0.7

[13N]-NH3 SUVmax RV/LV lateral wall

30.5

-4.4-65.5

0.08

RV ESV index/LV ESV index

0.9

-1.8-3.8

0.5

Comment 2: Another problem is the abundance of less relevant measurements such as lung function tests, which make the reading difficult. It is preferable to report on the measurements used to test your hypothesis. IPAH patients are known to have normal or near-normal lung function. In general, less relevant measurements can be simply summarized in the text.

Response: We agree that data on lung function test was abundant, as exclusion criteria were clearly defined and patients with respiratory comorbidity were not enrolled into the study. In fact, we deliberately selected IPAH patients without comorbidity for PET/CT study in order to exclude potential influence on investigated parameters.

We deleted lung function test data from Table 1

Comment 3: I have no objection to the use of national PH guidelines, but the references list should be revised and updated to integrate higher quality papers and be more tightly related to the study

Response:  We revised the reference list and integrated papers tightly related to the study.

Comment 4: Please define all abbreviations

Response: We defined all abbreviations in “Abbreviations” section.

Comment 6: Please comment more clearly on the added value of this study compared to your previous publications

Response: We have modified the “Conclusion” section:

Right ventricular [18F]-FDG myocardial uptake was an independent predictor of peak oxygen consumption in IPAH patients without comorbidities.

CPET has proven to be a reliable non-invasive tool for differentiation between IPAH patients without comorbidities with low-risk and those with intermediate-high risk status.

Accumulation of data on right ventricular myocardial metabolism in comparison with traditional invasive and non-invasive methods for PAH severity assessment requires prospective validation in a larger cohort of patients with IPAH.

Round 2

Reviewer 1 Report

Comments and Suggestions for Authors

This study expands on the link between right ventricular myocardial metabolism and idiopathic pulmonary arterial hypertension.While the topic is of potential interest,the work lacks sufficient convincing data to support its claims. In its current form,the manuscript does not meet the standards for publication and its therefore not recommended for acceptance.The following limitations remain:

  1. In Table 6, multivariate regression did not find any association between PET/CT parameters and VE/VCO2 in CPET. The study results are limited and lack clinical guidance significance.
  2. In previous review comments, we recommended that the authors use the 2022 ESC/ERS guideline diagnostic criteria as the inclusion criteria for IPAH patients. In their response, the authors explained that the guidelines had not yet been updated at the time of patient enrollment, but subsequent risk stratification referenced the 2022 ESC/ERS guideline criteria, resulting in inconsistency.

Reviewer 2 Report

Comments and Suggestions for Authors

I appreciate the authors' effort to improve their work. I am convinced by their point-by-point reply